# Distributed Group Coordination of Random Communication Constrained Cyber-Physical Systems Using Cloud Edge Computing

1st Hongru Ren
*School of Automation*
*Guangdong University of Technology*
Guangzhou, China
renhongru2019@gdut.edu.cn

2nd Yinren Long
*School of Automation*
*Guangdong University of Technology*
Guangzhou, China
2112204425@mail2.gdut.edu.cn

3rd Hui Ma
*School of Mathematics and Statistics*
*Guangdong University of Technology*
Guangzhou, China
huima2022@gdut.edu.cn

4th Hongyi Li
*College of Electronic and Information Engineering*
*Southwest University*
Chongqing, China
lihongyi2009@swu.edu.cn

*Abstract*—This long abstract studied the distributed group coordinated control problem of cyber-physical systems (CPSs) with multi-agent architecture. We build the distributed networked multi-group agent systems (NMGASs) with nonlinear and unknown dynamics via cloud edge computing. The common and challenging situations of random communication constraints in CPSs are considered, including network-induced delay, packet dropout, and packet disorder, which are treated as round-trip time (RTT) delay. To actively compensate for RTT delay and achieve coordination among all agents, a data-driven cloud edge predicted control strategy is designed. This strategy only needs to obtain the I/O measurement data of the systems, and can automatically carry out adaptive learning, which has more extensive application scenarios compared to model-based control methods. Theoretical analysis yields the conditions of simultaneous stability and consensus of the closed-loop systems with the proposed strategy. Finally, the practical examples are provided to illustrate the effectiveness of the proposed strategy.

*Index Terms*—Cyber-physical systems, cloud edge computing, data-driven control, networked predictive control, random communication constraints.

## I. Contribution

Our study aims to tackle two key questions. Firstly, we aim to establish a novel communication structure for distributed NMASs in mixed networks. Secondly, we propose a data-driven cloud edge predicted control scheme to overcome the challenges imposed by random communication constraints. The main contributions of this long abstract are summarized as follows.

1) The cloud edge computing is employed to construct the distributed networked multi-group agent systems (NMGASs). Agents are grouped based on network conditions, with each group being assigned to an edge node, and all edge nodes communicate with the cloud node. Parameter estimation is carried out in the edge nodes, while the control strategy is executed in the cloud node. Compared with [1] and [2], which only utilized cloud computing to establish distributed NMGASs, this approach simplifies the network topology, reduces the information interaction between agents, and enhances the system's capabilities for handling large volumes of real-time data.

2) Our study extends the method of compensating communication constraints proposed in [3] to distributed NMGASs. In this long abstract, a data-driven cloud edge predicted control strategy is proposed to address random communication constraints in each transmission channel of distributed NMGASs and achieve effective coordination of all agents.

3) The necessary and sufficient conditions of simultaneous consensus and stability of the distributed NMGASs with the data-driven cloud edge predicted control strategy are provided.

## II. Main Results

For the practical cyber-physical systems located in mixed networks, data exchange between agents poses a significant challenge. Thus, the distributed NMGASs is established by leveraging cloud edge computing. The structure of distributed NMGASs using cloud edge computing is shown in Fig. 1. Firstly, agents are categorised into different groups based on network conditions and every group of agents is allocated to a edge node. Then, all edge nodes are connected to a cloud node. Consequently, agents within a group can directly communicate with their respective edge nodes through the network, while all edge nodes collectively communicate with the cloud node.

Fig. 2 gives the output trajectories of each group of agents and the reference signal to be tracked.

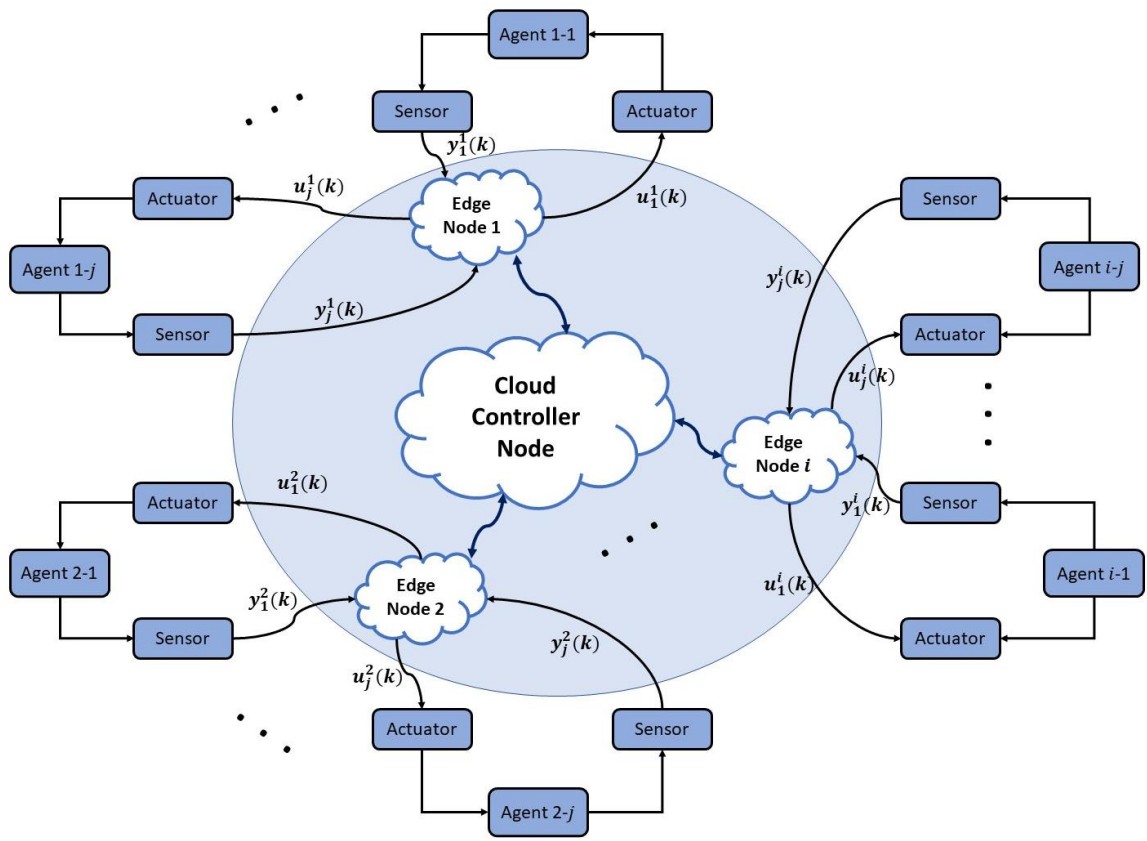

Fig. 1. Structure of distributed NMGASs using cloud edge computing.

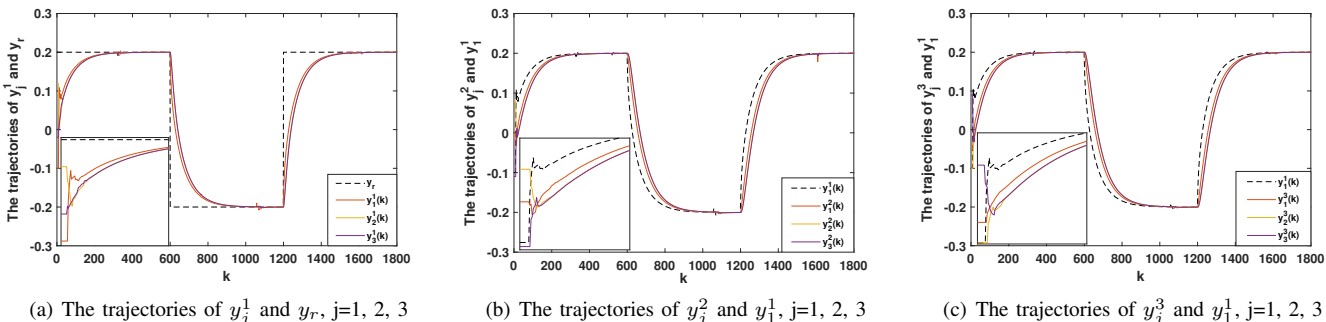

(a) The trajectories of $y_j^1$ and $y_r$, j=1, 2, 3

(b) The trajectories of $y_j^2$ and $y_1^1$, j=1, 2, 3

(c) The trajectories of $y_j^3$ and $y_1^1$, j=1, 2, 3

Fig. 2. Coordination of the distributed NMGAS with compensation.

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
