# OpenReview forum: "Distributed Group Coordination of Random Communication Constrained Cyber-Physical Systems Using Cloud Edge Computing"
_IEEE.org/ICIST/2024/Conference — IEEE ICIST 2024 Conference Submission_

### Official Review · Reviewer_rGB4 · 2024-08-21
**This long abstract studied the distributed group coordinated control problem of cyber-physical systems (CPSs) with multi-agent architecture. The feasibility of the designed control approach is proven via the simulation example. However, the following suggestions need to be considered in the revised manuscript to further improve the quality of this paper.**

**Rating:** 7
**Confidence:** 3

**Review:**

1. How does the proposed approach simplify the network topology compared to the methods in [1] and [2]?
2. In what ways does the reduction of information interaction between agents contribute to the handling of large volumes of real-time data?
3. Please carefully check the grammar errors in this paper to ensure readability and fluency for the readers.

---

### Official Review · Reviewer_wcn3 · 2024-08-22
**Review Results**

**Rating:** 6
**Confidence:** 4

**Review:**

This paper provides a study on the distributed group-coordinated control of cyber-physical systems with a multi-agent architecture, focusing on the challenges posed by random communication constraints such as network-induced delay, packet dropout and packet disorder.

1.How does the cloud edge computing infrastructure enhance the performance of distributed NMGASs in comparison to traditional centralized or purely cloud-based approaches?

2.Can the proposed strategy be applied to CPSs with varying scales and complexities, or does it have limitations that restrict its applicability?

3.How does the cloud edge predicted control strategy compare in terms of efficiency and robustness to other existing model-based control methods?

4.Please provide the complete framework of this paper.

---

### Official Review · Reviewer_YBJJ · 2024-08-22
**This paper studies the distributed group coordinated control problem of cyber-physical systems (CPSs) with multi-agent architecture. There are the following questions need to be considered:**

**Rating:** 6
**Confidence:** 4

**Review:**

This paper studies the distributed group coordinated control problem of cyber-physical systems (CPSs) with multi-agent architecture. There are the following questions need to be considered:
1. Can you elaborate on the fundamental principles of cloud edge computing and how it builds the distributed networked multi-group agent systems (NMGASs) with nonlinear and unknown dynamics？ 2. The idea proposed in this paper is novel, and it is suggested to enrich the content to facilitate readers' understanding.

---

### Decision · Program_Chairs · 2024-09-08

Accept (Oral)